# Choroidal Thickness in a Hyperopic Pediatric Population

**DOI:** 10.3390/diagnostics12102330

**Published:** 2022-09-27

**Authors:** Vanessa Antonia Gerena Arévalo, Jose Maria Ruiz-Moreno

**Affiliations:** 1Department of Ophthalmology, Puerta de Hierro-Majadahonda University Hospital, 28222 Madrid, Spain; 2Sanitas La Moraleja University Hospital, 28050 Madrid, Spain; 3Department of Ophthalmology, Castilla de la Mancha University, 13001 Ciudad Real, Spain

**Keywords:** hyperopia, swept-source optical coherence tomography, choroid, children

## Abstract

Aims: To evaluate the choroidal thickness (CT) in hyperopic and emmetropic children using swept-source optical coherence tomography (SS-OCT). Methods: This was a prospective, cross-sectional comparative study. Macular choroidal thickness and axial length of 62 eyes from hyperopic pediatric patients were studied. CT was determined at nine different macular locations. The results were compared to 66 eyes of healthy pediatric patients. Results: Study groups were classified as a hyperopic group (SE ≥ 2D) and an emmetropic group (SE < 2D). The hyperopic group have shorter AL than the emmetropic group (*p* < 0.001). The mean CT is greater in the hyperopic group (*p* = 0.039), and there are no significant differences between CT and gender (*p* = 0.389). Study participants were also classified by age (2–5 years old and 6–18 years old), and we observe differences in CT, but these differences are only significant for the 6–18 years old group (*p* < 0.05). Conclusions: CT in hyperopic pediatric populations is statistically thicker than in healthy pediatric patients. AL and SE have statistically significant correlations with CT values, and those correlations are seen in children in the ocular slow-growing phase (6–18 years old), and not in the early years (2–5 years old).

## 1. Introduction

Pathological hyperopia in childhood can lead to visual dysfunctions such as strabismus or amblyopia, and may be the origin of poor performance at school due to distraction, poor reading skills [1,2], or headaches. Previous research studied the associated factors of hyperopia, such as age, outdoor activities, axial length (AL), corneal radius [3,4,5,6], and genetics [7,8].

To evaluate the role of choroid variations in refractive status such as myopia and hyperopia, and after learning the essential role of the choroidal layer in ocular pathologies and its influence on the regulation of eye growth by remodeling the scleral extracellular matrix and controlling the defocus of the eye [9,10,11,12], new studies are using swept-source optical coherence tomography with spectral domain (SS-OCT) technology [13,14,15].

Prior studies investigated the choroidal thickness (CT) in general populations, concluding that there are correlations between the choroid and AL, age [16], and refractive status [3,17], but few studies compared mean or sub-foveal CT [13,16,18] in young hyperopic children without amblyopia.

In this cross-sectional study, we investigated choroidal characteristics using SS-OCT in children aged 4–17 years of age, evaluating choroidal morphological variations, comparing the total mean CT with age and AL, and differentiating between refractive status.

## 2. Materials and Methods

This cross-sectional study was performed according to the tenets of the Declaration of Helsinki, and was approved by the Ethic Committee of Hospital Puerta de Hierro- Majadahonda, Madrid, Spain (approval code PI 128/18).

Every parent and child understood the study protocol and provided written informed consent.

A sample of 128 eyes of 128 pediatric patients (4–17 years old), who met inclusion criteria: emmetropic (spherical equivalent (SE) 0-2D) or hyperopic (SE > 2D), was randomly collected in the ophthalmology consulting unit in Hospital Puerta de Hierro-Majadahonda and Hospital Sanitas La Moraleja. All the examinations were carried out in the afternoon to avoid diurnal variations in choroid thickness.

Exclusion criteria were amblyopia (anisometropia of 2 diopter (D) or greater), best-corrected visual acuity (BCVA) of less than 20/40, astigmatism of 2D or greater, intraocular surgery of pathology, any systemic disease, or any birth or gestation problem.

All patients underwent a complete ophthalmological examination, including BCVA using the Snellen test, sensorimotor examination, slit lamp biomicroscopy, cycloplegic refraction (with >6 mm of pupil dilation or non-reactive pupil using 3 drops of 1% cyclopentolate) (Colicursi Ciclopléjico, Alcon Co., Barcelona, Spain), and fundus examination.

For those patients who met the inclusion criteria, and after parental consent was obtained, the same investigator carried out complementary tests, and included non-contact optical biometry (IOL Master 700; Carl Zeiss Meditec, Jena, Germany) to measure the AL and swept-source optical coherence tomography with spectral domain Triton (SS-OCT), using the 12-line radial (12 mm surface area) scan pattern program to study the CT.

The SS-OCT software was used to segment retinal layers and construct topographic maps. All the acquired images were inspected, and if automatic segmentation errors occurred, manual segmentation was performed.

The CT was measured between the Bruch membrane and the scleral interface, and after a correct segmentation (Figure 1), the Early Treatment Diabetic Retinopathy Study (ETDRS) circular grid (most frequent scanning protocol used) [19] was applied to the tomography maps, which divided the macula into three concentric circles centered on the fovea: the central foveal circle (1 mm), parafoveal circle (3 mm), and perifoveal circle (6 mm). Then, this was divided into the superior, inferior, nasal, and temporal areas (Figure 2).

Refractive status groups (hyperopic and emmetropic) were classified using spherical equivalent (SE), and, using the meta-analysis conducted by Castagno et al. (2014) on hyperopia prevalence as a reference [3], the hyperopic group was defined as SE ≥ 2 diopters (D), and the emmetropic group as 0–<2 D.

In previous studies [5,17,20], a strong correlation between age and axial length is demonstrated, so, for our study, we classified patients by age and growth periods [21] to assess the correlation between axial length and choroid in each group:0–18 months old: the rapid postnatal phase where the axial length increases by 3.7–3.8 mm;2–5 years old: slower phase, decreased growth velocity (1.1–1.2 mm);6–18 years old: slow juvenile phase where the eye attains emmetropia.

### Statistical Analysis

The statistical package for the social sciences (SPSS) program was used for statistical analyses. The normality of the samples was evaluated using the Kolmogorov–Smirnov test. The variables in the hyperopic patients did not show a normal distribution. Thus, non-parametric statistics were used in this study. The Mann–Whitney U test was used to compare parameters between two groups (AL, CT, SE, age, gender). Statistical significance was defined as *p* < 0.05. The Spearman rho correlation coefficient was used to detect influential factors in CT. Finally, a multiple regression analysis was carried out with the significant variables.

## 3. Results

The baseline characteristics of the study population are presented in Table 1. The median age is 9 years (range: 4–17 years old); 43.8% are boys, and 56.3% are girls. In the hyperopic group, the median AL is 22.04 mm (interquartile range IQR 4.65) and the SE ranges between +2D and +9.25D. In the emmetropic group, the median AL is 22.97 mm IQR 3.5 and the SE ranges between 0 D and 1.88D (Table 1).

The hyperopic group shows a shorter AL than the emmetropic group, and has higher CT values in topographic measures than the emmetropic group (Figure 3).

As shown in Table 2, there are no significant correlations between mean CT and gender, and there is a non-significant direct correlation with age. Correlations between AL, SE, study groups, and mean CT show statistical significance.

All patients were separated into three groups based on age (0–18 months old, 2–5 years old, 6–18 years old) in order to analyze their CT and axial length. As there are no patients in the 0–18 months old group, we only have two groups.

As shown in Table 3 and Table 4, in the 2–5 years old age group, there is no clinical significance for CT and SE. The correlation between mean CT and AL is positive, but not significant (*p* = 0.305). In the 6–18 years old age group, there are significant correlations in CT values; the correlation between mean CT and AL is negative (−0.366) and clinically significant (*p* < 0.001).

Macular CT multiple regression analysis (Figure 4) including AL and SE and controlling for age has an adjusted R2 of 0.155 with high significance (*p* = 0.001). The SE is not statistically significant, and, thus, the equation to calculate mean CT is:Mean macular CT(6–17 y) = 630.633 − 15.464 (LA)

## 4. Discussion

Studies were previously carried out on the multifunctionality of the choroid, showing that, apart from being the vascular layer for the retina, it contributes to thermoregulation and production of cytokines and growth factors, modulates intraocular pressure by controlling blood flow, intervenes in the drainage of aqueous humor, and adjusts the position of the retina by modifying its thickness to regulate the defocus of the eye, increasing its size if there is myopic defocus or decreasing it if its defocus is hypermetropic [10,12,15]. Zhang et al. described that this process can be altered by drugs that inhibit nitric oxide, preventing the increase in thickness secondary to myopic defocus [9], which opens up possibilities for new therapies for refractive alterations that are modified by the choroid, such as myopia.

Due to these studies, and with the possibility of being able to study the choroid with new technology, such as SS-OCT, we carried out a cross-sectional study of the retina, overcoming the pigment epithelium and allowing us to obtain knowledge on the pathophysiology and etiology of some ocular medical conditions [9,22].

In this study, we observed the possible relationship and role of choroidal thickness with refractive status (hyperopic and emmetropic) in pediatric patients (4–17 years old). We assessed the correlation between the two variables to open the possibility of future therapies that modify the choroid and improve pathologic hyperopia (>2D) in children, as is performed for myopia [8].

Initially, observing the results of choroidal thickness in nine CT zones of all patients, the zone with the thickest CT is the temporal zone, followed by the superior zone, central fovea, and inferior zone, and the thinnest zone is the nasal zone, as shown in adults in previous studies [13,16,17]. Therefore, we can say that, with ocular growth, although the structures increase in size, the vascular proportions in the choroid remain the same as in adulthood [9,22].

On the other hand, a comparison of CT between study groups according to refractive status (hyperopic and emmetropic) shows that hyperopic children (with an SE of more than 2D) have greater mean choroidal thickness than emmetropic children. This is clinically relevant, as it may suggest that refractive status may be influenced, among other things, by vascular changes in the choroid.

The SE is the number of diopters of each eye and the cause for which a patient is classified in refractive states. Usually, 0D is an emmetropic eye, positive values are hyperopic and negative values are myopic, but in pediatric populations, emmetropia is considered as an hyperopic eye less than 2D [3]. Thus, we wanted to assess the correlation that SE has with choroidal thickness, avoiding classification bias and ensuring satisfactory and clinically relevant results.

Xiong et al. report that in children, there are no significant differences between choroidal thickness and gender, and describe that AL increases with growth and choroidal thickness decreases [22]. However, there are studies that do not associate choroidal thickness with age in the pediatric population [11,16].

Due to this, we analyzed the correlation between AL and CT, which shows a statistically significant correlation in these patients. However, AL presents a very close relationship with age, as is demonstrated in several studies [3,17,22]. There is a greater emphasis on AL in pediatric populations due to ocular development, where the eye is born hyperopic with a length of 22 mm and, in three growth phases, reaches its normal length of 23–24 mm, achieving the emmetropia that occurs in adolescence.

For this reason, in this study, data on the relationship between AL and CT were analyzed in each growth phase of the eye. In our study, patients were classified into two groups: one from 2 to 5 years old, and the other from 6 to 17 years old. The results show that in the 2–5 years old group, there is no correlation between CT and SE, or AL, while in the 6–17 years old group, there is a relationship between these variables, i.e., the shorter the AL, the thicker the CT.

This may be because, in healthy eyes, the initial development presents a rapid growth rate of all ocular structures, with probably no relationship between the variables studied. However, when growth slows down (6–18 years old), relevant factors that modify the CT can be observed.

The result is an inverse correlation between these variables, i.e., the longer the axial length, the thinner the choroidal thickness, which is consistent with previous clinical studies [9,13,16] where patients with longer AL, such as myopes, have thinner choroidal thickness.

An analysis of the SE variable and its relationship with CT in each age group was also performed, with a similar result to the AL variable, where in the younger age group (2–5 years old), there is no relationship between both variables, while for the 6–17 years old group, there is a positive correlation between them, where the greater the spherical equivalent (more hyperopic), the thicker the choroidal thickness, which coincides with the previous results of the groups in this study according to refractive status.

In other words, having a greater or positive spherical equivalent, or being hyperopic, leads to a thicker choroidal thickness than in those with a lower spherical equivalent, or who are emmetropic in this case. However, when the multivariate analysis was performed, controlling for age, spherical equivalent, unlike AL, does not present a significant result. Therefore, unlike adults [9,10], children’s SE is probably not an important influencing factor for CT, or vice versa. This is different from the nerve fiber layer, as, according to Banc’s systematic review of pediatric retinal parameters in OCT [13], retinal nerve fiber layer thickness is not influenced by age or gender, but does have a positive association with spherical equivalent.

The multivariate analysis of choroidal thickness according to axial length explains only 15% of the result. Therefore, although it is important to study the results according to age groups in children, it would also be necessary to study other ophthalmologic and sociocultural data, such as hours in front of screens or close-up tasks and hours of play time outside the home, among others, to elucidate other causes that may influence choroidal thickness.

A limitation in this study is the number of patients in the 2–5 years old group, since this is much smaller than the second group and, thus, differences may not be seen for this reason.

In future studies, it will be important to bear the age parameter in mind when assessing changes and influential factors in ocular structures in the pediatric population.

## Figures and Tables

**Figure 1 diagnostics-12-02330-f001:**
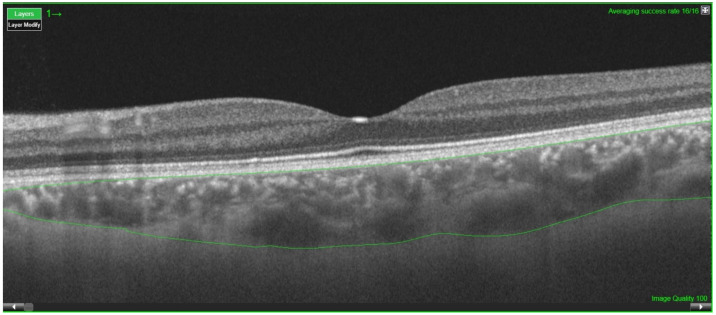
Macular OCT with 12-line radial scan and retina layers defined.

**Figure 2 diagnostics-12-02330-f002:**
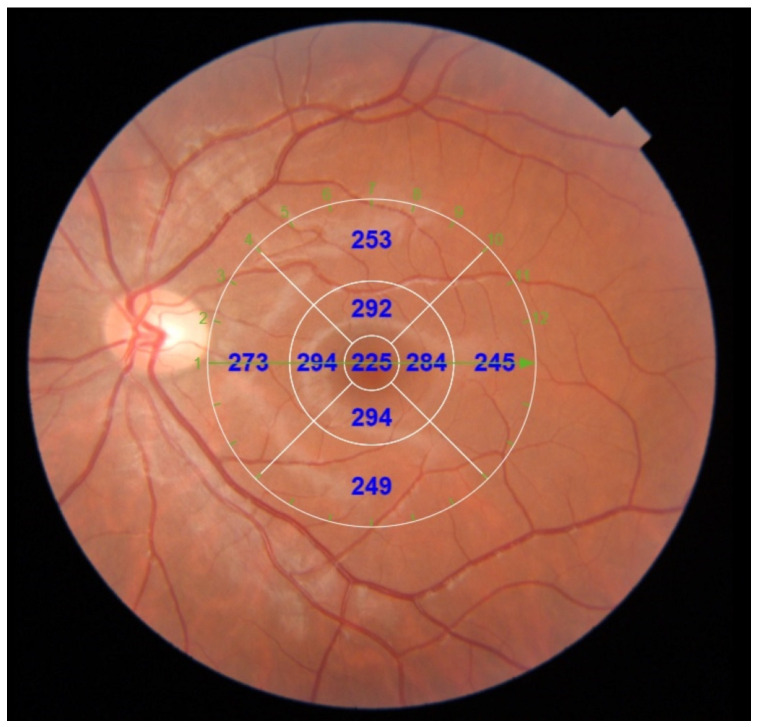
Macular OCT with EDTS grid, which divides the macula into three concentric circles centered on the fovea: the central foveal circle (1 mm), parafoveal circle (3 mm), and perifoveal circle (6 mm). This was divided into the superior, inferior, nasal, and temporal areas.

**Figure 3 diagnostics-12-02330-f003:**
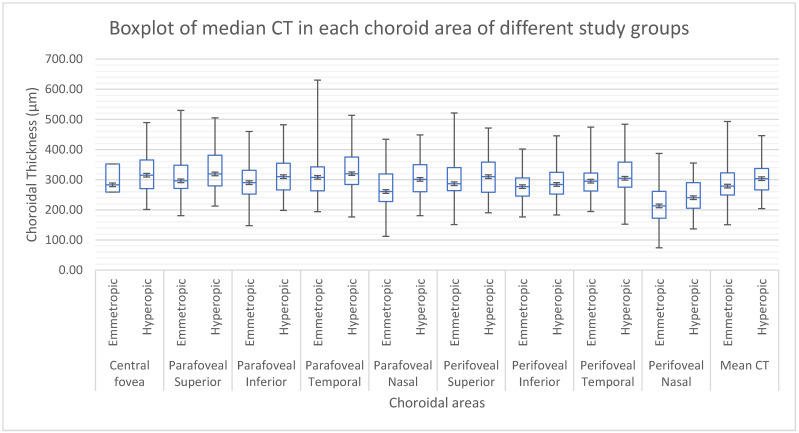
Median CT in each choroid area in different study groups.

**Figure 4 diagnostics-12-02330-f004:**
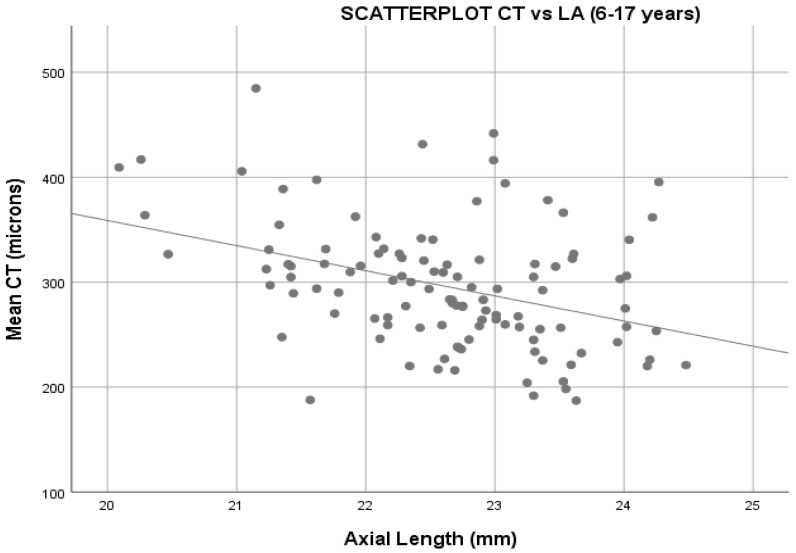
Scatterplot showing results for choroidal thickness and axial length in patients 6–17 years old.

**Table 1 diagnostics-12-02330-t001:** Baseline characteristics of study population.

Parameter	Hyperopic *n* = 62		Emmetropic *n* = 66		Mann–Whitney U
	Median	IQR	Median	IQR	
AGE	9	12	9.5	13	*p* = 0.682
AL	22.04	4.65	22.90	3.5	*p* < 0.001
SE	3.88	7.25	0.75	1.88	*p* < 0.001
MEAN CT	303.22	297	279	244.45	*p* = 0.039

AL: axial length; SE: spherical equivalent; CT: choroidal thickness; IQR: interquartile range.

**Table 2 diagnostics-12-02330-t002:** Correlations between biometric factors and mean CT in all patients.

		Mean CT	SEX	AGE	AL	SE
Correlation coefficients	STUDY GROUP	−0.183 *p = 0.039*	0.154 *p = 0.083*	0.036 *p = 0.684*	0.486 *p = < 0.001*	−0.866 *p = < 0.001*
	AGE	0.023 *p = 0.797*			0.276 *p = 0.002*	−0.049 *p = 0.582*
	AL	−0.298 *p = 0.001*				−0.617 *p = < 0.001*
	SE	0.261 *p = 0.003*				

AL: axial length; SE: spherical equivalent; CT: choroidal thickness.

**Table 3 diagnostics-12-02330-t003:** Correlations between biometric factors and mean CT (2–5 years old age group) *n* = 16.

		Mean CT
Correlation coefficients	AL	0.274 *p = 0.305*
	SE	0.018 *p = 0.948*

AL: axial length; SE: spherical equivalent; CT: choroidal thickness.

**Table 4 diagnostics-12-02330-t004:** Correlations between biometric factors and mean CT (6–17 years old age group) *n* = 112.

		Mean CT
Correlation coefficients	AL	−0.366 *p = < 0.001*
	SE	0.287 *p = 0.002*

AL: axial length; SE: spherical equivalent; CT: choroidal thickness.

## Data Availability

All the data generated or analyzed during this study are included in this article. Further enquiries can be directed to the corresponding author.

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
