# Peer review of "Choroidal Thickness in a Hyperopic Pediatric Population"

_diagnostics, 2022, doi:10.3390/diagnostics12102330_

Round 1

Reviewer 1 Report

Dear authors, the content of the paper is interesting, but there are several issues to amend.

Firstly, I strongly recommend authors to contact a language editor, since throughout the text there are many times that the authors mix the present with the past, and inappropriate expressions are often used (for example as in line 61 , "patients who HAD inclusion criteria--> patients who MET inclusion criteria).

Secondly, there are several statistical errors. Such as, from line 122 to128, where the authors show a correlations table, but they explain diferences about groups....Also there are several statements in discussion about data not provided previously in results. For example in line 184-187, where are the data of the 9 ETDRS quadrants of the macula? In the results they only talk about a CT, which, by the way, at no time they explain what it is. Also in line 214-218 the authors have not carried out, or do not provide, in the results the anova statistics that show such a conclusion.

Thirdly, the format of tables 2,3 and 4, is quite upgradeable. I recommend that each p value be under its correlation coefficient. 

Mean CT

sex

age

AL

SE

Sex

Pearson's

Coefficient

p

Also, for figure 4, I do not think an excel plot is the most appropriate, and a line plot do not represent the distribution of the data, please change it for a SPSS box plot.

Fourthly, statistic reasults are so simple, but based on disscussion, it seems that authors have performed more analysis than the provided in the results, but if is not on results should not be on discussion, so please improve your results and provide all data or remove it from discussion.

Finally, methodology is so weak. What BCVA test has authors used (Snellen, logMAR, LEA test..). Were amblyopic eyes excluded?, If yes, please provide the BCVA limit score. Also there are some concerns about the use of SE to classify, there was no astigmatism limit??Because a high astigmatism could have a high impact on SE and also affect OCT measurements due aberrations.

Reviewer 2 Report

This is a very interesting paper analyzing choroidal thickness in hyperopic children. However, there are some methodological issues to be explained.

The number of the Ethics Commitee approval should be given. Inclusion criteria are not given, only exclusion criteria. You wrote that only hyperopic and emmetropic eyes were included.

In the methods section it has been written as follows: A sample of 128 eyes of 128 pediatric patients (4-17 years) was randomly collected.

And later: For further analysis, patients were also classified by age by growth periods [21]: • 0-18 months: the rapid postnatal phase where the axial length increases of 3.7- 88 3.8mm • 2-5 years: slower phase, that decrease growth velocity (1.1-1.2mm) • 6-18 years: slow juvenile phase where the eye attains emmetropia.

There is inconsistence, please, correct. If the children were randomly selected, why where are no myopia cases?

Round 2

Reviewer 1 Report

Line 29:

bibliography format [3-7]

Line 33:

bibliography format [10-13]

Line 35:

bibliography format [14-16] Line53: 128 eyes from 128 patients. All eyes were included? All of them met the inclusion criteria?   Line 73-79: Triton provides 9 quadrants, but which of them has been used for statistics? In the results, Mean CT is cited, but where does this value come from? Does the device give it or has it been calculated?   Figures 1&2: I sugest autor to merge both images. Also to describe mor extensively figure 1 caption. Maybe describe that this image provides  the 9 cuadrants and the name of them.   Line 110-114:

Why have the authors used the median and IQR instead of the mean and SD? Although the distribution does not follow normality, it would have been more appropriate.

Suddenly on line 106, they provide a value of AL 22.97+0.99 which seems to be a Mean with their SD. Please reconcile, always use the Mean with the SD or always the median with the IQR.  

Figure 4:

It should be Figure 3, as there is no previous figure after figure 2.

Thanks for the new format, looks more apropriate. Have the authors calculated if there are differences between emetropyc and hyperopic patients in each of the 9 quadrants apart the Mean CT?   Table 2: Gender is a qualitative variable (male or female, 1 or 2), it cannot be correlated with Mean CT, age, AL or SE, it would be a violation of statistical laws. Please remove it from table 2. On the other hand, are the data in Table 2 from all the patients or only from the hyperopic patients? If so, please use the term "hyperopic patients"   Line 148: Again plase use the appropriate term, hyperopic patients.   Line 151-155: Tables 3&4 don’t show differences between groups, no statistics between hyperopic groups were performed. There was no clinical significance for CT between study groups” There’s no significant correlations between CT, AL and SE in the 2-5 years groups. Again in line 153,In the 6–18-years-old age group there were significant differences in CT values”,change for significant correlations.   Line 155: A correlation coefficient of -3.78 is impossible. Please review the data and correct if necessary, because in table 4 appears -0.366.   Table 3 & 4: Please provide N from each group. In table 4 and line 155 there is a mistake, “<0.01”, missing zero, should be <0.001 Also to merge both tables will be so practical.   Sometimes group 6-18, appears as 6-18 and sometimes as 6-17. Please reconcile.   Line 165-169: Multiple regresion analysis is so confusing to understand.It is just 3 line to explain a very complex concept. Also Mean CT correlation with SE is highly significant (p=0.002).  

Figure 5:

Should be Figure 4.

Discusion:

Line 200-202:

If there are no adults in the sample, how do the authors arrive at this statement? If it is obtained from previous studies, please provide a bibliography.

Line 208:

SE is a mathematical formula, not the number of diopters in each eye.

Line 224-234

Although the N of group 2-5 is low, it still missing to know if there are statistical differences between children aged 2-5 and those aged 6-18. Because line 226-227 appears “ The results showed that in the 2–5-years-old group, there were no differences in CT between groups…Where is this statistics? No statistics between hyperopic groups were performed, only separated correlations in each group.

Author Response

Response to Reviewer 1 Comments

Thank you very much for your comments. We believe that they have helped us in an important way to improve the quality of our article.

  1. Line 29: bibliography format [3-7], Line 33: bibliography format [10-13], Line 35: bibliography format [14-16].

Corrected

  1. Line53:128 eyes from 128 patients. All eyes were included? All of them met the inclusion criteria?

128 eyes were all included and met inclusion criteria.

  1. Line 73-79:Triton provides 9 quadrants, but which of them has been used for statistics? In the results, Mean CT is cited, but where does this value come from? Does the device give it or has it been calculated?

The 9 quadrants were used to obtain the choroidal thickness measurements given by OCT; Comparisons of each zone between groups (emmetropic and hyperopic) were also made.

The mean choroidal thickness value was the mean value of the 9 choroidal zones measurements in each eye, using the spss program. This was to assess the choroid a single value, and then correlate it with the variables under study. 

  1. Figures 1&2:I sugest autor to merge both images. Also to describe mor extensively figure 1 caption. Maybe describe that this image provides the 9 cuadrants and the name of them

I modify the order of the images and make the suggested changes, however I not merge them because the first one represents the segmentation of layers, in this case the choroid according to OCT and the second one represents the areas where the choroid was measured.

  1. Line 110-114:

Why have the authors used the median and IQR instead of the mean and SD? Although the distribution does not follow normality, it would have been more appropriate.

Thank you for your comment. Consulted with our statistician he advises to use the IQR in this work.

  1. Figure 4: It should be Figure 3, as there is no previous figure after figure 2.

Corrected

  1. Have the authors calculated if there are differences between emetropyc and hyperopic patients in each of the 9 quadrants apart the Mean CT? 

Yes, indeed the differences in the 9 quadrants were calculated giving differences in each quadrant by groups, being clinically significant, however the purpose was to measure the overall or mean choroidal thickness in each patient.

  1. Table 2: Gender is a qualitative variable (male or female, 1 or 2), it cannot be correlated with Mean CT, age, AL or SE, it would be a violation of statistical laws. Please remove it from table 2. On the other hand, are the data in Table 2 from all the patients or only from the hyperopic patients? If so, please use the term "hyperopic patients"  

Corrected , were all patients

  1. Line 148: Again plase use the appropriate term, hyperopic patients.   

Corrected, were all patients

  1. Line 151-155:Tables 3&4 don’t show differences between groups, no statistics between hyperopic groups were performed. “There was no clinical significance for CT between study groups” There’s no significant correlations between CT, AL and SE in the 2-5 years groups. 

Corrected, it was indeed a correlation with the spherical equivalent.

  1. Again in line 153,“In the 6–18-years-old age group there were significant differences in CT values”,change for significant correlations.   

Corrected

  1. Line 155:A correlation coefficient of -3.78 is impossible. Please review the data and correct if necessary, because in table 4 appears -0.366.   

Corrected

  1. Table 3 & 4:Please provide N from each group. 

Corrected

  1. In table 4 and line 155there is a mistake, “<0.01”, missing zero, should be <0.001 Also to merge both tables will be so practical.   Sometimes group 6-18, appears as 6-18 and sometimes as 6-17. Please reconcile.   

Corrected

  1. Line 165-169:Multiple regresion analysis is so confusing to understand. It is just 3 line to explain a very complex concept. Also Mean CT correlation with SE is highly significant (p=0.002).  

The multiple regression model was performed with the Mean CT variable as the dependent variable and the independent variables were those whose correlation with Mean CT was statistically significant (AL p< 0.001 and SE p= 0.002). However, when making the model, the variable SE was not statistically significant (p= 0.060), which means that it has no relevance when performing the analysis with the other variable.

The results obtained are explained using the adjusted R2 and statistical significance.

  1. Figure 5: Should be Figure 4.

Corrected

Discusion:

  1. Line 200-202: If there are no adults in the sample, how do the authors arrive at this statement? If it is obtained from previous studies, please provide a bibliography.

Corrected, bibliography [10,22]

  1. Line 208: SE is a mathematical formula, not the number of diopters in each eye.

SE value is obtained by a mathematical formula, using as data the sphere and the astigmatism, both expressed in diopters, so the unit of measurement of the spherical equivalent data is diopters (D).

  1. Line 224-234 Although the N of group 2-5 is low, it still missing to knowif there are statistical differences between children aged 2-5 and those aged 6-18. Because line 226-227 appears “ The results showed that in the 2–5-years-old group, there were no differences in CT between groups…Where is this statistics? No statistics between hyperopic groups were performed, only separated correlations in each group.  

Corrected, I understand the confusion. I took the variable SE to correlate with the variable CT of all patients avoiding the classification bias by groups (hyperopic and emmetropic), however it is true that I did not directly use the qualitative variable of study groups, so I have modified it.

The results showed that in the 2–5-years-old group, there were no correlation between CT and SE neither AL